# Divergent Regulation of Myotube Formation and Gene Expression by E2 and EPA during In-Vitro Differentiation of C2C12 Myoblasts

**DOI:** 10.3390/ijms21030745

**Published:** 2020-01-23

**Authors:** Orly Lacham-Kaplan, Donny M. Camera, John A. Hawley

**Affiliations:** 1Exercise and Nutrition Research Program, Mary Mackillop Institute for Health Research, Australian Catholic University, Melbourne 3000, Australia; john.hawley@acu.edu.au; 2Department of Health and Medical Sciences, Swinburne University of Technology, Melbourne 3122, Australia; dcamera@swin.edu.au

**Keywords:** 17β-estradiol, n-3PUFA, eicosapentaenoic acid, transcriptome, C2C12, myogenesis

## Abstract

Estrogen (E2) and polyunsaturated fatty acids (n-3PUFA) supplements independently support general wellbeing and enhance muscle regeneration in-vivo and myotube formation in-vitro. However, the combined effect of E2 and n-3PUFA on myoblast differentiation is not known. The purpose of the study was to identify whether E2 and n-3PUFA possess a synergistic effect on in-vitro myogenesis. Mouse C2C12 myoblasts, a reliable model to reiterate myogenic events in-vitro, were treated with 10nM E2 and 50μM eicosapentaenoic acid (EPA) independently or combined, for 0–24 h or 0–120 h during differentiation. Immunofluorescence, targeted qPCR and next generation sequencing (NGS) were used to characterize morphological changes and differential expression of key genes involved in the regulation of myogenesis and muscle function pathways. E2 increased estrogen receptor α (Erα) and the expression of the mitogen-activated protein kinase 11 (Mapk11) within 1 h of treatment and improved myoblast differentiation and myotube formation. A significant reduction (*p* < 0.001) in myotube formation and in the expression of myogenic regulatory factors Mrfs (*MyoD*, *Myog* and *Myh1*) and the myoblast fusion related gene, *Tmem8c*, was observed in the presence of EPA and the combined E2/EPA treatment. Additionally, EPA treatment at 48 h of differentiation inhibited the majority of genes associated with the myogenic and striated muscle contraction pathways. In conclusion, EPA and E2 had no synergistic effect on myotube formation in-vitro. Independently, EPA inhibited myoblast differentiation and overrides the stimulatory effect of E2 when used in combination with E2.

## 1. Introduction

Satellite cells (SCs), the quiescent adult muscle stem cells, are responsible for muscle hypotrophy and regeneration during postnatal development and adulthood [1,2]. For regeneration, activated SCs differentiate into myocytes (fusible myoblasts) and fuse with damaged myofibers [1]. At the cellular level, activation of tissue-specific transcription factors and increased gene abundance prompts the exit of dividing SCs from the cell cycle into the myogenic (myogenesis) pathway [2,3]. Interaction between transcription factors and signal transduction pathways results in the expression of numerous myogenic regulatory factors (Mrfs) that govern the normal progress of myogenesis [2,3].

A reduced number of SCs and an inability to undergo myogenesis may be the outcome of, or contribute to skeletal muscle disorders such as atrophy, cachexia and sarcopenia [4]. In this regard, it has been the aim of many scientists to optimize myogenesis in-vitro to improve therapies to alleviate symptoms associated with muscle atrophy and degeneration.

The results from several studies suggest that the reproductive hormone estrogen (E2) and n-3 polyunsaturated fatty acids (n-3PUFA) play favorable roles in maintaining skeletal muscle mass and function. For example, postmenopausal women undergoing estrogen-based hormone replacement therapy (HRT) to alleviate the symptoms associated with menopause have greater muscle mass and strength compared to women not undergoing HRT treatment [5,6,7,8]. Maintaining muscle function through nutritional strategies has also been an area of intense research [9] with considerable focus on fish oil. Fish oil contains n-3PUFA that are known to improve muscle strength, particularly in older women [10,11]. However, while some studies find supportive effects of E2 or n-3PUFA on myoblast differentiation in-vitro [12,13,14,15,16], others have shown an inhibition [17,18,19,20,21,22].

The cellular mechanism(s) activated by E2 and n-3PUFA regulating myogenesis are equivocal. MAPK/ERK and PI3k/Akt transduction pathways have been both positively and negatively implicated in regulating the transcription, translation and post-translation modifications of Mrfs to support myogenesis following E2 or fatty acid treatment [13,15,18,23]. Findings from several studies suggest a positive and synergistic effect of reproductive hormones and fish oil supplements on cardiovascular health of postmenopausal women [24,25,26]. However, their combined effect on muscle regeneration has not been investigated. In the present study we tested the hypothesis that E2 and n-3PUFA would induce a positive and synergistic effect on myogenesis in-vitro. From the numerous n-3PUFA investigated to date, eicosapentaenoic acid (EPA) has been found to have the most significant effect regulating myogenesis in-vitro [14,16,17,22]. Hence, we examined myogenesis fate in mouse C2C12 myoblasts treated with E2 and EPA independently or in combination, for 0–24 h or 0–120 h from induction of differentiation, with a focus on morphological changes and differential expression of key genes involved in muscle differentiation and muscle function pathways. 

## 2. Results

### 2.1. Morphological Changes and Myoblast Fusion Index

Morphological changes to C2C12 myoblasts cultured in control-vehicle (Con-Ve), E2, EPA and E2/EPA solutions were determined by immunofluorescence following treatment with anti-Desmin mouse antibody known to be expressed in both myoblasts and myotubes with increased expression as differentiation progresses [27]. When using immunofluorescent analyses, the staining is weak in the cytoplasm of undifferentiated myoblasts and gradually intensifies in elongated myocytes, early tubes and myotubes. Because of its structural distribution with myotube differentiation along the length of a muscle fiber and in close association with the plasma membrane and between myofibrils, Desmin staining defines the tube structure [28,29,30] making it a valid marker for early and mature myotubes.

In order to confirm the reliability of Desmin as a marker for in-vitro-derived C2C12 myotubes, in a separate experiment we treated the same cultures with Desmin and MYH antibodies at 120 h of differentiation using two different secondary fluorescent colors and showed that the staining overlapped (Figure 1), confirming that Desmin can be used as a marker for myotube development.

Morphological changes were recorded from the time differentiation was induced (0 h time), up to 120 h (Figure 2).

The highest number of elongated myoblasts at 48 h and the highest number of fully formed tubes at 120 h were both found in E2 treated cultures (Figure 2 and Figure 3). E2 treatment also resulted in the highest fusion index of all conditions and was significantly different compared with EPA and E2/EPA (*p* < 0.001, Figure 3). Cells treated with EPA or with E2/EPA had a significantly lower number of initial or mature myotubes (*p* < 0.01) and the lowest fusion index (*p* < 0.001) compared to Con-Ve and E2 treatments, respectively (Figure 2 and Figure 3).

The total number of nuclei within microscopic fields increased with time (*p* < 0.05) in all treatment groups with no differences between treatments at any given time point. Collectively, these morphological data indicate that exposure to E2 during myoblast differentiation improves myotube formation in-vitro while EPA and/or the combined E2/EPA treatment repressed formation of myotubes.

### 2.2. Gene Expression

#### 2.2.1. Time-Dependent Expression of Individual Genes Relative to 18S Ribosomal RNA Using Real Time qPCR Analysis

##### *Gpr30*, *Erα* and *Erβ*

Expression of the three estrogen receptors *Gpr30*, *Erα* and *Erβ* at 0–24 h and 0–120 h are presented in Figure 4.

Relative expression of *Erα* peaked at 1 h (1.33-fold) only in cells treated with E2 followed by a downregulation at 6 and 24 h.

The expression of *Gpr30* and *Erβ* did not change from 0–24 h. While the expression of *Erα* stayed low in all treatments at 6 and 120 h, the relative expression of *Gpr30* and *Erβ* increased significantly (*p* < 0.001) at 120 h in both Con-Ve and E2 treated cells, but not in cultures treated with EPA or E2/EPA (Figure 4c–f). Collectively, these results suggest that *Erα* expression in muscle stem cells is E2 dependent and occurs during early stages of myogenesis. In contrast, both *Gpr30′s* and *Erβ’s* expression was independent of E2 and was associated with fully established myotubes.

##### *Mapk11* and *Akt1*

*Mapk11* expression increased significantly (1.6 fold; *p* < 0.001) at 1 h only in E2 treated cells (Figure 5a,b). Following this initial increase, *Mapk11* expression decreased over time. *Akt1* expression did not change throughout the five days of culture for any treatment (Figure 5c,d).

##### *MyoD1*, *Myog*, *Myh1* and Myomaker (*Tmem8c*)

*MyoD1* expression decreased over time in all treatments between 0–24 h (Figure 6a; *p* < 0.001). Expression of *MyoD1*, *Myog* and *Myh1* increased at 120 h in Con-Ve and E2 (2–11 folds; *p* < 0.001) but did not change or decreased in EPA and E2/EPA treated cells (Figure 6b–d). The expression of *Tmem8c* increased over time in both Con-Ve and E2 treated cells (*p* < 0.05 at 48 h and *p* < 0.001 at 120 h) but not in EPA or E2/EPA treated cells (Figure 6e).

### 2.3. Next Generation Sequencing (NGS)

Differential expression analysis was completed on 12,987 from 26,586 genes identified in the *mus musculus* database. These genes had >10 reads in at least one of the three mRNA samples extracted from each treatment. Heatmaps and glimma plots of differentially expressed genes showed greater similarities between Con-Ve and E2 treated cells than cells treated with EPA (Figure 7).

In EPA treated cells, 6343 and 6248 genes had lower expression than Con-Ve and E2 treated cells, respectively, with 43.2% and 45.3% of the genes statistically different between the groups (*p* < 0.05). Totals of 6644 and 6739 genes had higher expression in EPA treated cells than in Con-Ve or E2 treated cells, respectively, with 39.5% and 42.4% of the genes statistically different between the groups (*p* < 0.05). A comparison between E2 and Con-Ve treatments showed that although 6491 genes had lower and 6497 had higher expression in E2 treated cells, only 8.9% and 7.6% genes were statistically different between the groups (*p* < 0.05), emphasizing similarities between the groups.

#### 2.3.1. Pathway Enrichment Analyses, Gene Ontology and Gene Functional Category in Cultures at 48 h Treatment with Con-Ve, E2 and EPA

The 10 most significantly enriched pathways in E2 and EPA at 48 h of treatment when compared to untreated cells or to each other are presented in Figure 8. From these, the striated muscle contraction pathway was the most significantly enriched in E2 treated cells compared to Con-Ve cells (Figure 8a; *p* < 0.001) or EPA treated cells (Figure 8f; *p* < 0.001). The pathways enriched in EPA treated cells compared to Con-Ve or E2 treated cells were associated with inflammation and immune responses (Figure 8c; *p* < 0.003). The RAF/MAPK was also enriched at 48 h in EPA treated cells compared to Con-Ve cells (Figure 8e; *p* < 0.001). Gene ontology (GO) and functional category (GFC) analysis showed that genes associated with muscle structure and function had reduced expression in cells treated with EPA (Appendix A).

#### 2.3.2. Gene Expression Profile in the Myogenic and the Striated Muscle Contraction Pathways at 48 h Treatment with Con-Ve, E2 and EPA

We characterized the myogenic and striated muscle contraction pathways as they are responsible for muscle differentiation, structure and function. From the 24 genes identified in the myogenic pathway, 17 (71%) had higher expression in E2 treated cells compared with Con-Ve (Figure 9a). From these, *Myod1*, *Cdc42*, *Cdh2*, *Mef2c* and *Myog* were statistically different (*p* < 0.04–0.007). Genes significantly repressed by EPA treatment were *Mef2a*, *Mef2c*, *Mef2d*, *Myog*, *TCF12*, *Cdh15*, *Cdh2*, *Cdc42*, *Spag9*, *Cdon*, *Tcf4*, *Me2*, *Ctnnal* and *Myod1* when compared with E2 (Figure 9c; *p* < 0.01–0.001). The number of reads for *Mapk11*, *Myf5 and Myf6* was significantly higher in EPA treated cells (Figure 9c; *p* < 0.01–0.001).

A total of 41 out of 44 (93%) genes within the striated muscle contraction pathway had higher expression in E2 treated cells compared to Con-Ve cells with 20 (45%) significantly different (*p* < 0.001–0.05) including the myosin heavy chain gene family: *Myh1*, *Myh3*, *Myh4*, *Myh6*, *Myh7* and *Myh8.* All 44 striated muscle contraction genes had lower expression in EPA than in E2 treated cells with 39 (89%) significantly different (*p* < 0.001–0.05), suggesting overall downregulation of the striated muscle contraction pathway after EPA treatment (Table 1).

#### 2.3.3. Genes Unique to E2 and EPA Treated Cells

We identified genes uniquely expressed in E2 and in EPA treated cells (Appendix A). Genes of interest from the genes expressed in E2 and not in EPA treated cells were: fibromodulin (*Fmod*), known to regulate myoblast differentiation by controlling calcium influx into the cells [31]; fibroblast growth factor binding protein 1 (*Fgfbp1*), secreted by muscle tissue to slow age-related degeneration [32]; Shisa family member 2 (*Shisa2*), known to participate in myoblast fusion [33], fox-1 homolog (*Rbfox1*), known to promote the transcription factor myocyte enhancer factor 2D (*Mef2D*) splicing and subsequent myogenesis [34] and *Mef2D* itself.

Genes unique to EPA treated cells were mostly associated with immune response. Other genes in this list included: orosomucoid 1 (*orm1*), known to increase muscle glycogen [35]; endothelial PAS domain protein 1 (*Epas1*), known to promote adipose differentiation [36]; and lipocalin 2 (*Lcn2*), which regulates muscle regeneration [37]. Of interest is also haptoglobin (*Hp*), known to regulate adipose tissue development and fat metabolism [38]; and interleukin 6 (*Il6*), known to promote muscle differentiation and hypertrophy except when prolonged high doses are used [39,40].

## 3. Discussion

Several studies have reported that E2 and n3-PUFA, independently, have a positive effect on muscle function particularly in older women, and have stimulatory effects on myotube formation in-vitro. However, their combined effect on muscle regeneration in-vivo or in-vitro is unknown. Our study is the first to report that E2 and EPA induce divergent outcomes on myoblast differentiation, and that they have no synergistic effect when used in combination. Sequential imaging over five days of treatment with 10 nM E2, a commonly used concentration, increased myotube number and fusion index. However, both significantly decreased when cells were treated with a concentration of 50 μM EPA alone or combined with 10 nM E2 (Figure 3).

The present study focused on gene expression analyzed by qPCR and NGS, and while protein expression would have provided additional information regarding post-transcriptional changes, the morphological changes necessary for myoblast commitment to the myogenic lineage and their progress in myogenesis, are largely dependent on a complex network of genes [1,2]. We show that EPA interferes with this network by repressing or maintaining expression of specific genes typically regulated during differentiation, resulting in an attenuated myogenic pathway (Figure 9). For example, the reduced expression of *MyoD1* and *Myog* and increased expression of *Myf6* (also known as *Mrf4*) and *Myf5* in cells treated with EPA for 48 h suggests a lack of cellular commitment to the myogenic lineage (Figure 9) [2]. *Mapk11* was also expressed higher in these cells compared to Con-Ve or E2 treated cells (Figure 8). Mapk11 supports myoblast proliferation and initiation of myogenesis and has been shown to be downregulated after activating MyoD and MEF2C in committed myoblasts [41,42]. Even if EPA treated cells were committed to the myogenic lineage, they had limited ability to fuse with each other to form tubes as EPA repressed the expression of *Tmem8c* (Figure 6) and *Sisha 2* (Appendix A), which are essential for myoblasts’ membrane fusion and myotube extension [33,43]. EPA treatment also repressed the expression of genes associated with muscle function such as the *Myh* gene family, linked to muscle fiber type and substrate metabolism (Table 1) [44].

E2 activates cellular pathways via its intracellular receptors, Erα and Erβ, resulting in the receptors’ translocation to the nucleus to act as transcription factors that regulate E2-dependent gene expression [45]. E2 also interacts with a membrane G-protein coupled receptor 30 (GPR30) [32]. Both the intracellular and membrane receptors have been associated with in-vitro myoblast differentiation through signal transduction pathways [13,23,46]. Elevated expression of *Erβ* and *Gpr30* was independent of E2 treatment and was limited to cultures with fully formed myotubes in both Con-Ve and E2 treated cells. While the precise role of *Erβ* in fully formed myotubes is unknown, an increase in GPR30 during late myogenesis is attributed to its protective role from oxidative stress through activation of creatine kinase [46]. We observed a 1.3-fold increase in *Erα* expression at 1 h only in E2 treated cells followed by a downregulation. Although statistically not different to the Con-Ve group, we speculate that this peak in *Erα* expression is of biologically relevance and is E2 dependent. Ronda et al. [47] showed that the use of an antagonist of ERs and specific siRNAs to block *Erα* and *Erβ* expression resulted in Erα, but not Erβ, mediating ERK2 activation by 17β-estradiol.

Our findings show that *Mapk11*, and not *Akt1*, is the primary target to differentiation stimuli by E2. While *Akt1* expression remained unchanged through differentiation for all treatment groups, we observed a significant increase in the expression of *Mapk11* at 1 h with E2 treatment, supporting previous suggestions of regulation by the MAPK transduction cascade (Figure 5) [13,15,23,48]. The increase in *Mapk11* expression also coincided with an increase *Erα* expression, suggesting that the regulation of myogenesis through activation of the MAPK transduction pathway may be via this E2 receptor.

Cotreatment of C2C12 cells with E2 was insufficient to suppress EPA’s negative effect on myogenesis. This dominant effect of EPA has been reported in previous work on E2-dependent tumours where n-3PUFA prevented cell proliferation by altering lipid composition of the plasma membrane and subsequently interfered with E2-dependent signaling cascades such as the MAPK/Erk pathway [49]. Moreover, higher n-3PUFA to E2 ratios were shown to have a greater inhibitory effect on tumour cell proliferation [49]. Thus, the inability of E2 to overturn EPA’s inhibitory effect in the present study may be related to the high EPA to E2 ratio (5000:1).

Our finding that n-3PUFA (i.e., EPA) inhibits myogenesis is in agreement with accumulating evidence showing the negative effect of n-3PUFA on myotube formation in-vitro [17,19,22]. These studies highlight the potential negative effects of n-3PUFA consumption during pregnancy when de novo myogenesis is initiated at the embryonic and neonatal stages because disruption to myogenesis at this time may lead to postnatal muscle deficiencies in mass and function [50]. However, the n-3PUFA to E2 ratio is reduced considerably during pregnancy, thus preventing any proposed inhibitory effect of n-3PUFA on muscle development in-utero. It is estimated that levels of circulating E2 are nearly 100 times higher [51,52] while n-3PUFA in the blood are approximately 10 times lower [53] during pregnancy. Indeed, n-3PUFA ingestion has been associated with positive effects on newborn visual and cognitive development [54] and had no detrimental effects on postnatal development or in follow-up years after birth [55].

In contrast to our original research hypothesis, cotreatment of E2 and n-3PUFA did not have a synergistic effect on in-vitro myogenesis, raising the possibility that dietary supplements of fish oil (i.e., n-3PUFA) may interfere with the positive affect estrogen has on muscle regeneration in women on HRT. Recently, Ghnaimawi et al. [17] showed that treatment of differentiating C2C12 myoblasts with 50 μM EPA and DHA for four days increased adipogenesis and inflammatory-related genes and reduced tube formation. We did not examine adipose cell formation in our cultures following treatment with EPA, but were able to identify that *IL6* and adipose tissue related genes were uniquely expressed in cells treated with EPA (Appendix A). High doses of n-3PUFA intake have shown to increase the expression of uterine *Il6* [56].

In the present study we characterized morphological and molecular changes of myoblasts treated with E2 and EPA independently or combined. EPA significantly impaired the expression of muscle differentiation genes and genes associated with muscle function within 48 h of treatment. In contrast, E2 improved myoblast fusion and myotube formation and enhanced the expression of genes within the striated muscle contraction pathway. This indicates that reproductive hormone can be used to stimulate myoblast entry into the myogenic pathway for therapeutic purposes.

Further investigation is required to explore if n-3PUFA ingestion will increase intramuscular adipose tissue and whether ingestion while under HRT may override the positive effect E2 has on skeletal muscle. Further studies are also required to identify the optimal E2 to n-3PUFA ratio when consumed together in order to maximize the beneficial effect on general wellbeing with no harmful effect on muscle health.

## 4. Materials and Methods

### 4.1. Cell Culture

C2C12 murine myoblasts at passage 7–9 [57] (LONZA, Sydney, Australia) were cultured in proliferation medium composed of 4.5 g/L glucose Dulbecco’s modified eagle medium (DMEM; Life Technologies, Melbourne, Australia) supplemented with 10% fetal bovine serum (FBS; Sigma, Melbourne, Australia) and 0.2% penicillin/streptomycin (PenStrep; Life Technologies). Cells were plated onto 1% Geltrex (Life Technologies) coated 96-well dishes in proliferation medium at a density of 30,000 cells/mL (6000 cells/well). Culture dishes were maintained at 37 °C in a humidified incubator at 5% CO_2_/air mixture for 48 h to reach 80%–90% confluence. At 80%–90% confluence, proliferation medium was replaced with differentiation medium (4.5 g/L glucose DMEM and 0.2% Pen/Strep) containing 2% horse serum (HS; Sigma).

### 4.2. E2 and EPA Treatments

EPA and E2 were purchased from Sigma-Merck (Sydney, Australia). The company has certified EPA’s 1 3C NMR identity conformed to structure. EPA was used within six months of purchase. Frozen (−80 °C) stock solutions of 10 M E2 and 100 mM EPA (Sigma) in 100% ethanol were diluted into concentrations of 100 µM and 20 nM, respectively. Solutions were placed in a 56 °C water bath until EPA was dissolved, followed by further 1–2 h incubation at room temperature to allow EPA/BSA and E2/BSA conjugation. Solutions were then diluted in 4.5 g/L DMEM containing 2% HS to a final working concentration of 50 µM EPA and 10 nM E2. These concentrations were chosen based on previously published reports [13,22,58]. A control-vehicle (Con-Ve) solution contained the same proportion of 2% bovine serum albumin (BSA), 2% horse serum (HS) and ethanol (0.1%) and was handled like the treatment groups.

### 4.3. Immunofluorescence, Mytube Formation and Fusion Parameters

In an independent experiment we have shown Desmin to overlap with MYH expression in in-vitro derived C2C12 myotubes at 120 h of differentiation (Figure 1). For this, confluent cells (80–90%) were cultured in Con-Ve for 120 h. At 120 h, cultures were washed, fixed and analyzed by immunofluorescence as previously described [29,59]. Fixed cultures were stained with the appropriate secondary antibodies (Alexa Fluor 488; green and 594; red, Life Technologies) and costained with 4′,6-diamidino-2-phenylindole (DAPI) (Life Technologies) following an overnight incubation with anti-mouse Desmin (Life Technologies) and anti-mouse skeletal muscle myosin heavy chain antibody (Life Technologies). For the time-dependent development studies, cultures were treated only with anti-Desmin followed by the appropriate secondary antibody. At 0, 48 and 120 h, cultures were washed, fixed and analysed by immunofluorescence as before.

Stained cultures were visualised under the EVOS-II imaging system using the appropriate fluorescent filters. Pictographs were taken under 20× magnification with five image fields (one in the centre of the well and four others around the centre) from each treatment at 0 (myoblasts before differentiation), 48 (elongating myoblasts and myocytes and initiation of mytubes) and 120 h (fully formed myotubes; Figure 1). The number of elongated Desmin-positive myoblasts with ≥1 DAPI stained nucleus was recorded at 48 h. At 120 h, the number of Desmin-positive myotubes containing ≥2 DAPI stained nuclei, the total number of DAPI stained nuclei within the field and the total number of DAPI stained nuclei within tubes were all recorded. Fusion index was calculated as the proportion of nuclei within tubes from the total number of nuclei in the field.

### 4.4. Transcriptome Analyses

#### 4.4.1. RNA Extraction

Total RNA was extracted from cells using the PureLink RNA Mini Kit (Qiagen, Melbourne, Australia) following manufacturer’s instructions and stored at −80 °C until used for sequencing and target qPCR analyses.

#### 4.4.2. Real Time Quantitative PCR

Taqman-FAM-labelled primer/probes for the different genes cDNA was synthesized by using oligo (dT)20, and SuperScript^™^ III Reverse Transcriptase (Life Technologies, Australia) according to the manufacturers’ instructions. Time-dependent qPCR of differential expression was used to identify hierarchy of gene expression activated by E2 or by EPA treatment. We examined expression of the E2 receptors alpha and beta (*Erα: Esrra* Mm00433143_ml, *Erβ*: Esrrb Mm00442411_ml) and G protein-coupled estrogen receptor-30 (*Gpr30* Mm02620446_ml), known to initiate transduction cascades involved in myogenesis; the downstream target genes of these receptors, mitogen-activated protein kinase 11 (*Mapk11* Mm004440955_ml) and serine/threonine kinase 1 (*Akt1* Mm00437443_ml), known to activate Mrfs’ transcription and post-translation modification; the Mrf genes, myoblast determination protein (*MyoD* Mm00440387_ml), myogenin (*Myog* Mm00446194_ml), myosin heavy chain 1 (*Myh1* Mm01332489_ml); and the mouse myoblast fusion gene, myomaker (*Tmem8c* Mm00481256_ml), involved in myoblast fusion, myotube extension and myotube maturation.

Expressions of *Erα*, *Erβ*, *Gpr30*, *Mapk11*, *Akt1* and *MyoD* were measured at 0, 1, 6 and 24 h from initiation of differentiation. In a separate experiment, the expression of *Erα*, *Erβ*, *Gpr30*, *Mapk11*, *Akt1*, *MyoD* and *Myog*, *Myh1* and *Tmem8c* was examined at 0, 48 and 120 h. 18S ribosomal RNA (*Rn18s* Mm03928990_ml) was used as housekeeping gene for the early and late gene expression experiments.

RT-qPCR samples were mixed with Taqman probes in a final reaction volume of 20 µL. PCR protocol included 2 min at 50 °C for UNG activation, 10 min at 95 °C followed by 40 cycles of 95 °C for 15 s and 60 °C for 60 s. Relative amounts of mRNAs to Rn18s were calculated as ∆∆CT from three independent repeats with duplicates (total *n* = 6).

#### 4.4.3. Next Generation Sequencing (NGS) Using Illumina HiSeq 2500 RNA-seq

Differential gene expression was performed on mRNA extracted from Con-Ve, 10 nM E2 or 50 μM EPA treated cells at 48 h of differentiation. Transcripts were quantified, normalized and aligned against the *Mus musculus* genome database. The number of reads that were mapped to each known gene was summarized. A differential expression of genes (DEG) was used with "edgeR" (in R 3.5.0) to generate Log_2_FC (fold change) and significant differences (<0.05) represented by P and false discovery rate (FDR) for paired analyses of E2 vs. Con-Ve, EPA vs. Con-Ve and EPA vs. E2.

Gene ontology (GO) and gene functional category (GFC) analyses of upregulated and downregulated genes with Log_2_FC > 0.5 were performed using DAVID Bioinformatic (https://david.ncifcrf.gov) [60] followed by Pathway Enrichment (PE) analyses in Reactome (https://reactome.org) [61].

### 4.5. Statistical Analyses

All analyses within the study contained a total of three repeats with two replicates for each of the treatment groups (total *n* = 6). One- and two-way ANOVA were used to identify statistical differences associated with time and treatment, with *p* < 0.05 indicating significant differences for cell number, tube formation, fusion index and gene expression using the Holm–Sidak as post hoc test.

## Figures and Tables

**Figure 1 ijms-21-00745-f001:**
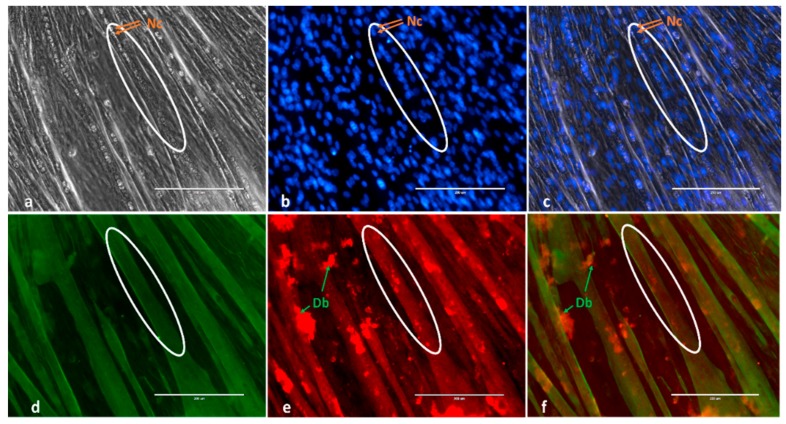
Example of myotube immunofluorescence staining and overlap of Desmin and MYH staining at 120 h in-vitro derived C2C12 myotubes. (**a**) Phase-contrast 4′,6-diamidino-2-phenylindole (**b**) (DAPI), blue; (**d**) Desmin, green; (**e**) MYH, red; (**c**) Merged Phase and DAPI and (**f**) Merged Desmin and MYH. Nc, Nuclei; Db, debris. Images were taken by the EVOSII imaging system (Thermo Fisher) at ×20. Circles mark the same tube presented in phase-contrast and Desmin and MYH markers.

**Figure 2 ijms-21-00745-f002:**
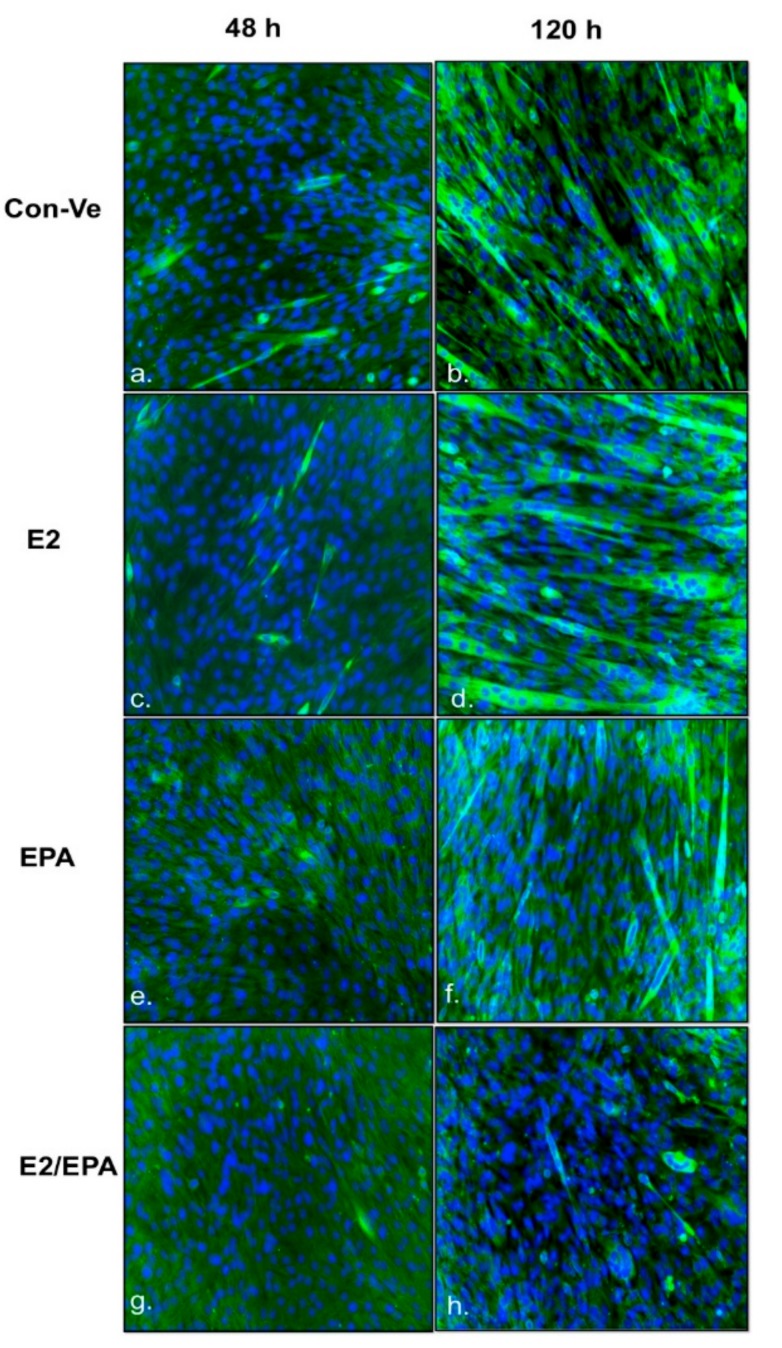
Myotube immunofluorescence staining of C2C12 treated with control-vehicle (Con-Ve) (**a**,**b**); 17β-estradiol (E2) (**c**,**d**); eicosapentaenoic acid (EPA) (**e**,**f**) and E2/EPA (**g**,**h**), at 48 and 120 h. Cultures were treated with anti-mouse Desmin followed by GFP secondary antibody (green) and nuclear DAPI (blue) staining. Images were taken by the EVOSII imaging system (Thermo Fisher) at ×20.

**Figure 3 ijms-21-00745-f003:**
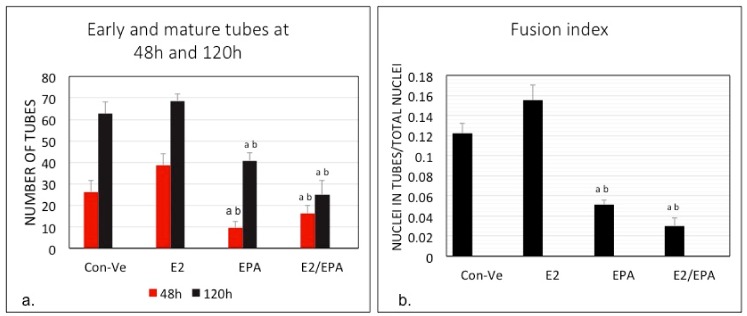
Number of tubes in five microscopic fields at 48 and 120 h (**a**) and fusion index (**b**) following treatment of C2C12 with Con-Ve, E2, EPA and E2/EPA. Myotube formation and fusion index were higher in E2 treated cells than the other groups. (*n* = 3; ^a^
*p* < 0.01; ^b^
*p* < 0.001 between EPA or E2/EPA to control and E2 at the same time point) (±SEM).

**Figure 4 ijms-21-00745-f004:**
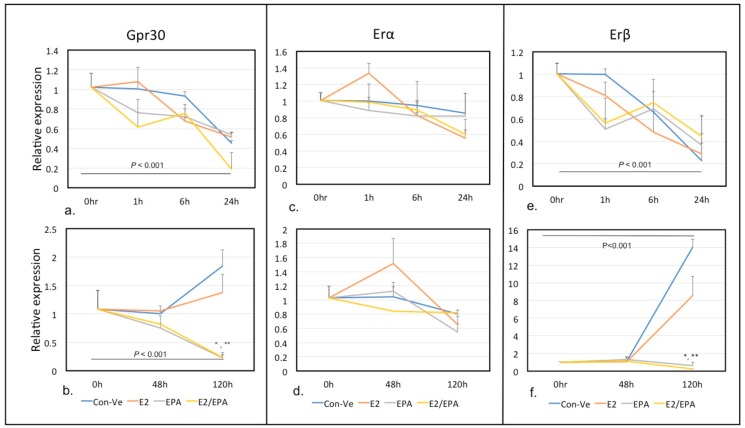
Time-dependent expression of E2 receptors *Gpr30*, *Erα and Erβ.* The expression of *Gpr30* and *Erβ* did not change at 0–24 h in all groups (**a**,**e**). *Erα* expression peaked at 1 h only in E2 treated cells followed by a decrease in expression at 6 and 24 h (**c**). The expression of *Erα* remained low for all other time points (**d**). The expression of *Gpr30* and *Erβ* increased significantly at 120 h in both Con-Ve and E2 treated cells (**b**,**f**). (*, ** *p* < 0.001 compared to Con-Ve at the same time point). (±SEM).

**Figure 5 ijms-21-00745-f005:**
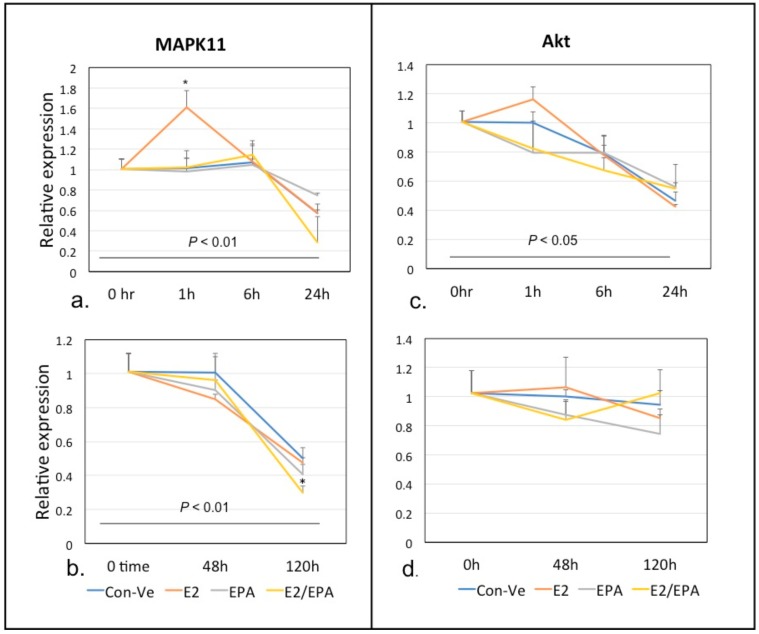
Time-dependent expression of *Mapk11* and *Akt1* at 0–24 h and 0–120 h. *Mapk11* expression peaked at 1 h only in E2 treated cells followed by a decrease in expression at all other time points (**a**,**b**). The expression of *Akt1* was similar in all groups (**c**,**d**). (* *p* < 0.001 compared to Con-Ve at the same time point) (±SEM).

**Figure 6 ijms-21-00745-f006:**
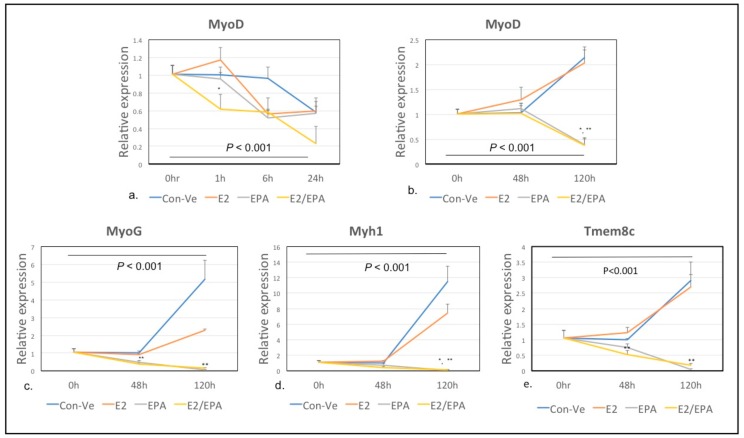
Time-dependent expression of *MyoD* at 0–24 h and 0–120 h (**a**,**b**), *myogenin* (**c**), *Myh1* (**d**) and *Tmem8c* (myomaker, **e**) at 0–120 h. *Myod* expression peaked slightly at 1 h in E2 treated cells. The expression of *MyoD*, myogenin and *Myh1* and *Tmem8c* increased in Con-Ve and E2 treated cells, and at 120 h of treatment with no change in EPA or E2/EPA treated cells (* *p* < 0.01; ** *p* < 0.001 compared to Con-Ve at the same time point; ±SEM).

**Figure 7 ijms-21-00745-f007:**
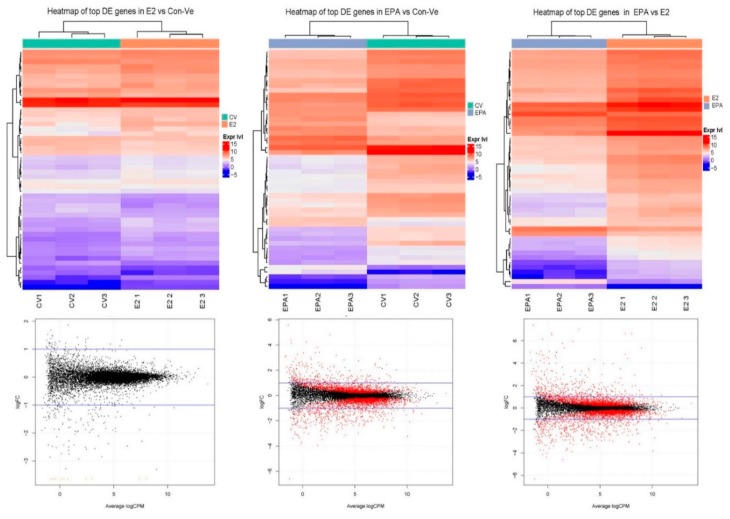
Paired analyses of heatmaps of the top 50 differentially expressed genes (DEG) and glimma plots of genes following Illumina HiSeq 2500 RNA-seq. Differences were greater between E2 and EPA treated cells than with Con-Ve. Con-Ve, green; E2, orange; EPA, light purple. Expression levels are marked from lowest (blue) to highest (red). Red dots in glimma plots represent significant differences (≥0.05) in false discovery rate (FDR) values.

**Figure 8 ijms-21-00745-f008:**
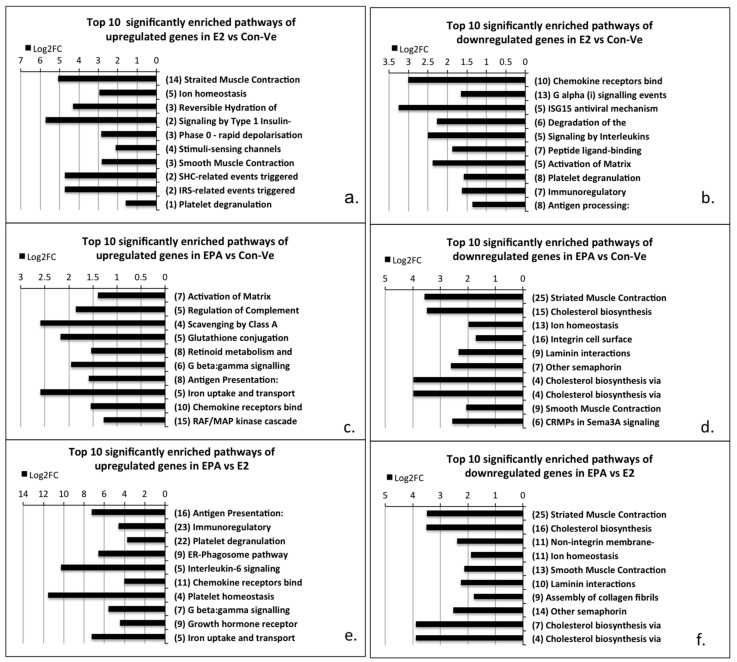
The 10 most significantly enriched pathways for upregulated (**a**,**c**,**e**)- and downregulated (**b**,**d**,**f**) genes with Log_2_FC > 0.5 comparing E2 vs. Con-Ve (**a**,**b**), EPA vs. Con-Ve (**c**,**d**) and EPA vs. E2 (**e**,**f**). Pathway enrichment analyses were performed using DAVID Bioinformatic (https://david.ncifcrf.gov) followed by identification in Reactome (https://reactome.org).

**Figure 9 ijms-21-00745-f009:**
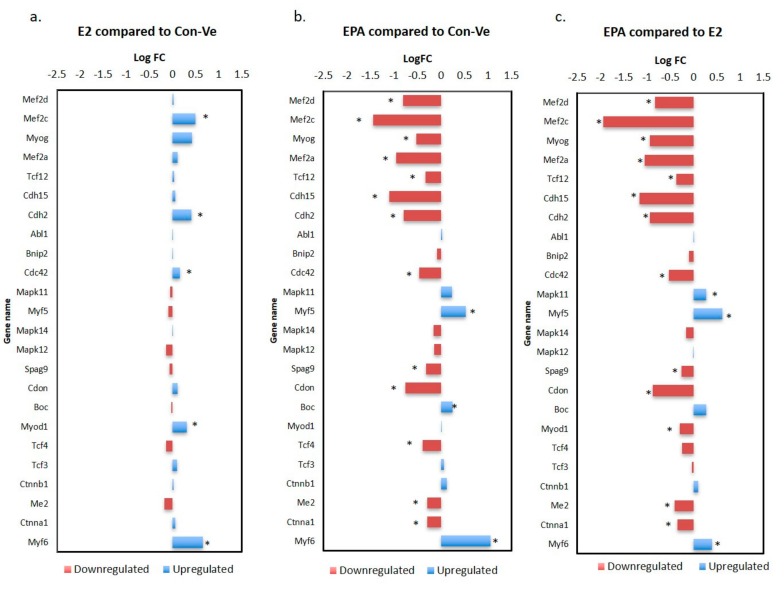
Paired analyses of differentially expressed genes in the myogenic pathway. The majority of the myogenic pathway genes were not differently expressed in E2 treated cells compared to Con-Ve (**a**). Similar genes in EPA treated cells had significantly lower expression compared to Con-Ve (**b**) or E2 treated cells (**c**) (* *p* < 0.05–0.001 between treatments).

**Table 1 ijms-21-00745-t001:** Expression profile of striated muscle contraction pathway genes.

Gene Name		E2 vs. Con-Ve			EPA vs. Cont-Ve			EPA vs. E2		
		logFC	*p* Value	FDR	logFC	*p* Value	FDR	logFC	*p* Value	FDR
*Acta1*	*actin, alpha 1, skeletal muscle*	0.56	0.048	0.606	−2.11	1 × 10^5^	0.0004	−2.67	1 × 10^6^	0.0001
*Acta2*	*actin, alpha 2, smooth muscle, aorta*	0.35	0.034	0.566	−0.328	0.045	0.1138	−0.68	0.0008	0.006
*Actc1*	*actin, alpha, cardiac muscle 1*	0.66	0.024	0.551	−2.53	2 × 10^6^	0.0001	−3.19	4 × 10^7^	4 × 10^5^
*Actg1*	*actin, gamma, cytoplasmic 1*	−0.02	0.831	0.985	−0.49	0.0005	0.005	−0.47	0.0006	0.005
*Actn2*	*actinin alpha 2*	0.92	0.001	0.364	−2.29	4 × 10^6^	0.0002	−3.22	1 × 10^7^	2 × 10^5^
*Actn3*	*actinin alpha 3*	0.24	0.029	0.557	−1.10	6 × 10^6^	9 × 10^5^	−1.35	1 × 10^7^	2 × 10^5^
*Actn4*	*actinin alpha 4*	0.05	0.352	0.886	−0.36	0.0001	0.002	−0.42	3 × 10^5^	0.0007
*Casq2*	*calsequestrin 2*	0.53	0.034	0.566	−2.43	1 × 10^6^	0.0001	−2.97	2 × 10^7^	3 × 10^5^
*Des*	*Desmin*	0.03	0.705	0.977	−0.52	5 × 10^5^	0.001	−0.55	10 × 10^3^	0.0007
*Jsrp1*	*junctional sarcoplasmic reticulum protein 1*	0.17	0.325	0.877	−0.69	0.001	0.014	−0.86	0.0004	0.003
*Mybpc1*	*myosin binding protein C, slow-type*	1.16	0.0002	0.226	−1.05	0.001	0.013	−2.21	3 × 10^6^	0.0001
*Mybpc2*	*myosin binding protein C, fast-type*	0.67	0.128	0.740	−2.23	0.002	0.017	−2.91	0.0003	0.003
*Myh1*	*myosin, heavy polypeptide 1, skeletal muscle, adult*	0.95	0.0006	0.300	−2.00	2 × 10^2^	0.0001	−2.96	9 × 10^9^	2 × 10^5^
*Myh3*	*myosin, heavy polypeptide 3, skeletal muscle, embryonic*	0.60	0.016	0.529	−3.21	7 × 10^8^	3 × 10^5^	−3.828	1 × 10^8^	1 × 10^5^
*Myh4*	*myosin, heavy polypeptide 4, skeletal muscle*	0.99	6 × 10^5^	0.183	−0.55	0.005	0.029	−1.55	1 × 10^6^	0.0001
*Myh6*	*myosin, heavy polypeptide 6, cardiac muscle, alpha*	1.10	0.0001	0.183	−1.69	2 × 10^5^	0.0008	−2.79	2 × 10^7^	3 × 10^5^
*Myh7*	*myosin, heavy polypeptide 7, cardiac muscle, beta*	0.93	0.0004	0.262	−1.34	0.0005	0.006	−2.15	8 × 10^6^	0.0002
*Myh8*	*myosin, heavy polypeptide 8, skeletal muscle, perinatal*	1.18	0.0006	0.295	−2.21	1 × 10^5^	0.0004	−3.39	2 × 10^7^	3 × 10^5^
*Myl1*	*myosin, light polypeptide 1*	0.50	0.035	0.566	−0.83	0.002	0.016	−1.34	8 × 10^5^	0.001
*Myl4*	*myosin, light polypeptide 4*	0.61	0.047	0.604	−1.74	0.0001	0.002	−2.36	1 × 10^5^	0.0003
*Myl9*	*myosin, light polypeptide 9, regulatory*	0.09	0.661	0.971	−0.90	0.002	0.016	−1.00	0.001	0.008
*Myom1*	*myomesin 1*	0.52	0.021	0.539	−2.00	2 × 10^6^	0.0001	−2.52	3 × 10^7^	4 × 10^5^
*Myom2*	*myomesin 2*	0.17	0.138	0.754	−0.86	1 × 10^5^	0.0005	−1.03	2 × 10^6^	0.0001
*Neb*	*Nebulin*	0.40	0.036	0.566	−2.02	4 × 10^7^	8 × 10^5^	−2.42	9 × 10^8^	2 × 10^5^
*Smpx*	*small muscle protein, X-linked*	0.49	0.047	0.604	−1.67	4 × 10^5^	0.001	−2.16	4 × 10^6^	0.0002
*Tcap*	*titin-cap*	0.87	0.031	0.561	−0.59	0.169	0.285	−1.47	0.003	0.015
*Tmod1*	*tropomodulin 1*	0.54	0.008	0.482	−0.84	0.0005	0.005	−1.38	1 × 10^5^	0.0003
*Tnnc1*	*troponin C, cardiac/slow skeletal*	0.51	0.060	0.624	−1.43	0.0002	0.003	−1.94	1 × 10^5^	0.0004
*Tnnc2*	*troponin C2, fast*	0.51	0.072	0.646	−1.72	7 × 10^5^	0.001	−2.23	9 × 10^6^	0.0003
*Tnni1*	*troponin I, skeletal, slow 1*	0.49	0.068	0.639	−1.90	2 × 10^5^	0.0007	−2.40	3 × 10^6^	0.0001
*Tnni2*	*troponin I, skeletal, fast 2*	0.39	0.187	0.800	−1.65	0.0001	0.003	−2.05	3 × 10^5^	0.0007
*Tnni3*	*troponin I, cardiac 3*	0.27	0.457	0.918	−0.90	0.05	0.129	−1.18	0.015	0.050
*Tnnt1*	*troponin T1, skeletal, slow*	0.32	0.146	0.764	−1.31	0.0001	0.002	−1.64	1 × 10^5^	0.0004
*Tnnt2*	*troponin T2, cardiac*	0.29	0.268	0.855	−1.97	2 × 10^5^	0.0007	−2.27	7 × 10^6^	0.0002
*Tnnt3*	*troponin T3, skeletal, fast*	0.39	0.152	0.769	−1.63	0.0001	0.001	−2.025	1 × 10^5^	0.0004
*Tpm1*	*tropomyosin 1, alpha*	0.13	0.139	0.756	−0.36	0.001	0.009	−0.49	0.0001	0.001
*Tpm2*	*tropomyosin 2, beta*	0.33	0.061	0.629	−0.68	0.001	0.012	−1.02	8 × 10^5^	0.001
*Tpm3*	*tropomyosin 3, gamma*	−0.09	0.342	0.885	−0.42	0.001	0.009	−0.33	0.005	0.023
*Tpm4*	*tropomyosin 4*	−0.08	0.299	0.868	−0.14	0.072	0.157	−0.06	0.38	0.506
*Ttn*	*titin*	0.19	0.152	0.769	−1.17	4 × 10^6^	0.0002	−1.37	1 × 10^6^	7E-05
*Vim*	*vimetin*	0.09	0.419	0.904	−0.03	0.728	0.810	−0.13	0.258	0.381

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
