# Peer review of "Divergent Regulation of Myotube Formation and Gene Expression by E2 and EPA during In-Vitro Differentiation of C2C12 Myoblasts"

_ijms, 2020, doi:10.3390/ijms21030745_

Round 1

Reviewer 1 Report

Lacham-Kaplan et al. is a fairly well written manuscript that describes the effect of E2 and/or EPA on skeletal myogenesis in vitro using C2C12 murine skeletal myotubes. The authors specifically target their analysis on the effects of E2 and EPA, however it would interesting to know whether they considered also using testosterone and other n-3 PUFA, specifically DHA and lenolenic acid, to compare the effects of E2 and EPA to other sex hormones and n-3 PUFA.

Specific Comments:

The authors need to do a better job of describing their decision to focus on EPA rather than other n-3 PUFA. Why is EPA more relevant than DHA to post-menopausal women, which is a focus of the authors reasoning in the introduction. Also, the authors mention in the Discussion that n-3 PUFA are suggested for increased consumption by expectant mothers. However, most prenatal supplements are primarily enriched with DHA. Plus, EPA can further desaturated and elongated to form DHA in by the liver in vivo. Therefore, dietary supplementation of EPA may not equate to increased in vitro exposure of EPA. The authors need to reconcile this discrepancy by further explaining the reason for their focus on EPA.

Figure 2: The design is a complete mess and needs to be redesigned to allow for proper interpretation of the data. The figure is difficult to interpret. The authors are encouraged to redesign and even expand the figure to allow the reader to examine the fluorescent images in a time-dependent manner. 

Figure 3: The use of superscripts to denote significant difference is rather confusing in this figure. Significant differences between experimental groups is difficult to determine, especially in panel B.

Figures 4: Panels A and B lack the name of the gene (GRP30) that they report, which needs to be fixed. Also, the authors should consider combining the panels A and B, C and D, E and F into single panels for each gene rather than two panels for each gene. The 0-24 hour graphs don't appear to add any significant information that could not be added to the 0-120 hour graphs. Perhaps a 0, 24, 48, and 120 hour X-axis would be more appropriate and would allow for a more streamlined figure that provides the same information in a more cohesive form. The current design seems redundant.

Figure 5: See above comments regarding Fig 4.

Discussion: As previously stated, the authors discuss the relevance of their data to dietary intake of n-3 PUFA in pregnant women. Again, DHA is the primary n-3 PUFA included in prenatal vitamins and much of the EPA is used to make DHA by the liver in vivo. Therefore, human embryos may not be exposed to excess levels of EPA in vivo. The authors need to address this describing why they focused on EPA and how their results can be properly interpreted regarding in vivo skeletal myogenesis.

Author Response

We appreciate the comments and useful suggestions made by the reviewer.

Comment 1:

Lacham-Kaplan et al. is a fairly well written manuscript that describes the effect of E2 and/or EPA on skeletal myogenesis in vitro using C2C12 murine skeletal myotubes. The authors specifically target their analysis on the effects of E2 and EPA, however it would interesting to know whether they considered also using testosterone and other n-3 PUFA, specifically DHA and lenolenic acid, to compare the effects of E2 and EPA to other sex hormones and n-3 PUFA.

Reply to comment 1:

As correctly indicated by the reviewer, the current study focuses on E2 and EPA in an in-vitro model of myogenesis. In our current research program we have been targeting ongoing studies primarily aimed at identifying the effects of n-3 PUFA (EPA, DHA and DPA) and reproductive hormones on muscle regeneration and muscle function. Thus, the suggestion by the reviewer on other reproductive hormones in combination with n-3PUFA is relevant and of consideration in our future studies.

Specific Comments:

Comment 2:

The authors need to do a better job of describing their decision to focus on EPA rather than other n-3 PUFA.

Reply to comment 2:

EPA and DHA have both been shown to regulate myoblast differentiation in-vitro with negative and positive effects on cell proliferation and/or tube formation, independently and combined. However, EPA has been more frequently studied and found to have a greater magnitude of effects than DHA on myotube formation (for negative effects at similar conditions and concentrations see: Hsueh et a., 2018; Zhang, J., et al, 2019; Ghnaimawi et al., 2019. For positive effects see: Megee et al., 2008; Lee et al., 2016; Saini et al., 2017). We have included this explanation in our introduction to further explain the use of EPA in (Lines 61-62).

Comment 3:

Why is EPA more relevant than DHA to post-menopausal women, which is a focus of the authors reasoning in the introduction.

Reply to comment 3:

We apologise for the confusion. We referred to post-menopause as a reference point for our study as in real life it is the only situation were E2 and n-3PUFA (DHA or EPA) will be co-ingested. EPA and DHA are both common ingredients in Fish-Oil supplements and at most times at similar concentrations. We have chosen EPA as per the reply to comment 1 rather than it having any significant effect in women at menopause compared to other n-3PUFAs.

Comment 4:

Also, the authors mention in the Discussion that n-3 PUFA are suggested for increased consumption by expectant mothers. However, most prenatal supplements are primarily enriched with DHA. Plus, EPA can further desaturated and elongated to form DHA in by the liver in vivo. Therefore, dietary supplementation of EPA may not equate to increased in vitro exposure of EPA. The authors need to reconcile this discrepancy by further explaining the reason for their focus on EPA.

Reply to comment 4:

We agree with the reviewer. We mentioned the concerns by other published reports, which referred to consumption of EPA or DHA during pregnancy merely to present our opposite view. In the discussion we suggest that the concerns made previously in other published reports are overemphasised as it is well documented that ingestion of n-3PUFA positively influences fetal and postnatal development. However, we do suggest that consumption should be with caution (like any other supplement during pregnancy).

We have modified the appropriate paragraph in the discussion L279/288 to be clearer with our views.

Comments and replies on figures

Figure 2: The design is a complete mess and needs to be redesigned to allow for proper interpretation of the data. The figure is difficult to interpret. The authors are encouraged to redesign and even expand the figure to allow the reader to examine the fluorescent images in a time-dependent manner.

In agreeance with the reviewer, we have now enlarged Figure 2 so that fluorescent images are presented in a time-dependent manner.

Figure 3: The use of superscripts to denote significant difference is rather confusing in this figure. Significant differences between experimental groups is difficult to determine, especially in panel B.

Figure 3 has been modified to clear significant differences

Figures 4: Panels A and B lack the name of the gene (GRP30) that they report, which needs to be fixed. Also, the authors should consider combining the panels A and B, C and D, E and F into single panels for each gene rather than two panels for each gene. The 0-24 hour graphs don't appear to add any significant information that could not be added to the 0-120 hour graphs. Perhaps a 0, 24, 48, and 120 hour X-axis would be more appropriate and would allow for a more streamlined figure that provides the same information in a more cohesive form. The current design seems redundant.

Figure 5: See above comments regarding Fig 4.

We acknowledge the reviewer’s comments Figure 4 and figure 5: have been modified to best present the data. The time line represent 0-24 (1 h, 6 h 12 h and 24 h) and 0-120 h (24 h intervals) are two parallel experiments aimed at showing immediate changes in gene expression within the first 24 h of differentiation and longer term changes up to 5 days. The expression is relevant to the 0 time point of the Con-Ve group in each experiment. Hence, we have presented them separately. During the first 24 h we see marked increase in the expression of Mapk and Erα relevant to the regulation of the myogenic pathway, whereas, for other genes expression markedly increased only at 120 h.

Comment 5:

Discussion: As previously stated, the authors discuss the relevance of their data to dietary intake of n-3 PUFA in pregnant women. Again, DHA is the primary n-3 PUFA included in prenatal vitamins and much of the EPA is used to make DHA by the liver in vivo. Therefore, human embryos may not be exposed to excess levels of EPA in vivo. The authors need to address this describing why they focused on EPA and how their results can be properly interpreted regarding in vivo skeletal myogenesis.

Reply to comment 5:

We appreciate the reviewer’s sentiment here. As addressed in comments 2 and comment 4, EPA and DHA have both been shown to regulate myoblast differentiation in-vitro with negative or positive effects on cell proliferation and/or tube formation independently or combined. However, most published studies have focused on EPA found it to have a more significant effect than DHA on myotube formation under similar conditions and concentrations (for negative effects see: Hsueh et a., 2018; Zhang, J., et al, 2019; Ghnaimawi et al., 2019. For positive effects see: Megee et al., 2008; Lee et al., 2016; Saini et al., 2017). We have included this explanation in our introduction to further explain the use of EPA. We mentioned the concerns by other published reports, which referred to consumption of EPA or DHA during pregnancy merely to present our opposite view. In the discussion we suggest that the concerns previously made in other published reports may be overemphasised as it is well documented that ingestion of n-3PUFA positively effect fetal and postnatal development. However, we do suggest that consumption should be with caution.

As mentioned above, we have modified the appropriate paragraphs in the discussion to be clearer with our views.

Reviewer 2 Report

The current work by Orly Lacham-Kaplan et al aimed at understanding the effect of E2 and EPA during myoblast differentiation is interesting.

However, there are few comments that need attention and would be an added information to the current findings.

For MAPKII and Akt, is it not possible to show the protein level expression for treatment times between 1-24h and for MyoD, Myogenin and MYH1, an immunofluorescence panel at 24h and 120h post-treatment would also be interesting to see. In Figure 6, the transcript levels of MyoD, Myogenin, and Myh1 are seen higher in the vehicle-treated samples in comparison to the E2 treated samples, however, Figure 2 shows that the myoblast differentiation is higher in E2 treated samples in comparison to the Vehicle control, how do you explain this difference in the results? None of your gene analysis data or RNA seq data mentions about Pax7, was Pax7 expression not detected in the differentially expressed gene sets?

Other minor comments are

Figure 2, Rearrange the figures in such a way that the 48h and120h are aligned as top and bottom panel, making it clearer. The 0h control can be presented as an inset in the figure. In figure 4, some of the figure labels are missing.

Author Response

We appreciate the comments and useful suggestions made by the reviewer.

The current work by Orly Lacham-Kaplan et al aimed at understanding the effect of E2 and EPA during myoblast differentiation is interesting.

However, there are few comments that need attention and would be an added information to the current findings.

Comment 1:

For MAPKII and Akt, is it not possible to show the protein level expression for treatment times between 1-24h and for MyoD, Myogenin and MYH1, an immunofluorescence panel at 24h and 120h post-treatment would also be interesting to see.

Reply to comment 1:

We agree with the comment made by the reviewer that expression of proteins in addition to genes is of interest. Additionally, this information may clear the issues raised by the reviewer in the next comment. However, this study focuses on gene expression (PCR and NGS) in C2C12 myoblasts and how it is affected by E2 and EPA treatment during differentiation.

We have modified the text in the discussion to reflect on the above (L234/237)

Comment 2:

In Figure 6, the transcript levels of MyoD, Myogenin, and Myh1 are seen higher in the vehicle-treated samples in comparison to the E2 treated samples, however, Figure 2 shows that the myoblast differentiation is higher in E2 treated samples in comparison to the Vehicle control, how do you explain this difference in the results?

Reply to comment 2:

Following from comment 1, the reviewer correctly indicated that transcript levels of the Mrfs MyoD, Myogenin and Myh1 at 120 h were higher in Con-Ve cells than E2 treated cultures, but the tube formation was higher in cultures treated with E2 at 120 h. It is important to note that no significant differences were identified in gene expression in both methods or in tube formation between the Con-Ve and E2 group at 48 h or 120 h. Additionally, although critical to differentiation, Mrfs are not the only players responsible for the formation of tubes. Although less transcripts are produced in E2 treated cultures than Con-Ve at 120 h, this may not ultimately represent the level of the protein as alluded to by the reviewer.

Comment 3:

None of your gene analysis data or RNA seq data mentions about Pax7, was Pax7 expression not detected in the differentially expressed gene sets?

Reply to comment 3:

The expression of Pax7 was not examined by qPCR as we tested only Mrfs associated with mid and late stages of myogenesis.

Pax7 was detected in cultures at 48 h analysed by NGS, with relatively low number of reads 224; 279 and 183 for Con-Ve, E2 and EPA, respectively. Although higher in E2 and lower in EPA treated cultures, no statistical differences between the groups. Were identified We did not include this information in the manuscript, as it did not impact on the outcome.

Other minor comments are

Figure 2, Rearrange the figures in such a way that the 48h and120h are aligned as top and bottom panel, making it clearer. The 0h control can be presented as an inset in the figure. In figure 4, some of the figure labels are missing.

Reply to minor comments:

We thank the reviewer for this suggestion. Figure 2 has now been modified and the 0 h control picture removed.

Reviewer 3 Report

Minor:

l 245: the Myh genes are not "responsible" for fibre type and "normal" muscle contraction, instead they mark fibre type and contribute to the type of contraction.

Major:

The quality of EPA should be checked i.e. if the double bounds are still in omega-3 position.

It must be stated that EPA was used in a comparable dose to those applied in other experiments ( which produced contradictory results).

Author Response

We appreciate the comments and useful suggestions made by the reviewer.

Minor:

l 245: the Myh genes are not "responsible" for fibre type and "normal" muscle contraction, instead they mark fibre type and contribute to the type of contraction.

Reply to minor comments:

We thank the reviewer for the clarification. We have corrected this in the manuscript L248/249

Major:

Reply to major comments:

Comment 1:

The quality of EPA should be checked i.e. if the double bounds are still in omega-3 position.

Reply to comment 1:

EPA was commercially obtained from Sigma-Merck (# E2011). The company guarantee it conforms to structure identity proton NMR spectra and > 99 % purity (GC) with a use period of >3 years.

Comment 2:

It must be stated that EPA was used in a comparable dose to those applied in other experiments (which produced contradictory results).

Reply to comment 2:

This information is provided in the material and method section 4.2. E2 and EPA treatments (L325/325) and also in the discussion (L230/233).

Round 2

Reviewer 1 Report

The authors of Lacham-Kaplan appear to have adequately addressed all of my concerns and comments. These revisions have greatly improved the quality of the manuscript. Therefore, I now believe that their manuscript should be accepted for publication. 

Author Response

We thank the reviewer for the positive comments

Reviewer 2 Report

I have gone through the response to the reviewer and agree with your explanation for the lack of protein-based data and Pax-7 expression.

Author Response

We thank the reviewer for accepting our explanation and the positive comments

Reviewer 3 Report

Concernig the manuscript I think the authors did not respond properly to the major criticism, if they have checked the purity of EPA. PUFAs are highly vulnarable to lipid peroxidation which causes oxidative stress. i.e. Birben et al, (2012)World Allergy Organ J. 5:9-19; Raedorstaff et al. (2015) Br. J. Nutr. 184:115-1122. Since the authors neglect this aspect the paper can not be accepted in its prese nt form.

Author Response

We thank the reviewer for the comment

Reply to comment 1

We agree with the comment that PUFAs are sensitive to peroxidation. Therefore n-3PUFAs obtained from Sigma-Merck only following certified testing for 1 3C NMR identity . Sigma-Merck has supplied the attached (below) certificate of analyses and recommendation of re-analyses after 3 years. We have been using all our n-3PUFAs and many other products from Sigma-Merck and find the products reliable. This information was added in L319/320.The use of EPA purchased never exceeded 6 months and was prepared as per the method described in Section 4 of the manuscript.

Round 3

Reviewer 3 Report

Thank you for supplementing the certificate of EPA purity. I think it would be worthwhie to insert the statement about the Sigma- Merck guarantee into the materials section. Otherwise to my view the paper is sufficient for publication.

Author Response

We thank the reviewer for the comment.

The details regarding the certified test for EPA structure is now mentioned in the method section 4.2 L319/321

The change is marked in yellow